# Investigating the role of the relaxin-3/RXFP3 system in neuropsychiatric disorders and metabolic phenotypes: A candidate gene approach

Win Lee Edwin Wong[1,2,3]*, Ryan Arathimos[4], Cathryn M. Lewis[4,5‡], Allan H. Young[3,6‡], Gavin S. Dawe[1,2,7‡]

1 Department of Pharmacology, Yong Loo Lin School of Medicine, National University of Singapore, Singapore, Singapore, 2 Healthy Longevity Translational Research Programme, Yong Loo Lin School of Medicine, National University of Singapore, Singapore, Singapore, 3 Department of Psychological Medicine, Institute of Psychiatry, Psychology & Neuroscience, King's College London, London, United Kingdom, 4 Institute of Psychiatry, Psychology and Neuroscience, Social, Genetic and Developmental Psychiatry Centre, King's College London, London, United Kingdom, 5 Faculty of Life Sciences and Medicine, Department of Medical and Molecular Genetics, King's College London, London, United Kingdom, 6 South London & Maudsley NHS Foundation Trust, Bethlem Royal Hospital, London, United Kingdom, 7 Life Sciences Institute, Neurobiology Programme, National University of Singapore, Singapore, Singapore

‡ These authors contributed equally to this work as senior authors
* win_lee_edwin.wong@kcl.ac.uk, edwin.wong@u.nus.edu

**Data Availability Statement:** UK Biobank data can be obtained from the UK Biobank project site, with further information available at (https://www.ukbiobank.ac.uk/ and http://biobank.ndph.ox.ac.uk/

## Abstract

The relaxin-3/RXFP3 system has been implicated in the modulation of depressive- and anxiety-like behaviour in the animal literature; however, there is a lack of human studies investigating this signalling system. We seek to bridge this gap by leveraging the large UK Biobank study to retrospectively assess genetic risk variants linked with this neuropeptidergic system. Specifically, we conducted a candidate gene study in the UK Biobank to test for potential associations between a set of functional, candidate single nucleotide polymorphisms (SNPs) pertinent to relaxin-3 signalling, determined using *in silico* tools, and several outcomes, including depression, atypical depression, anxiety and metabolic syndrome. For each outcome, we used several rigorously defined phenotypes, culminating in subsample sizes ranging from 85,881 to 386,769 participants. Across all outcomes, there were no associations between any candidate SNP and any outcome phenotype, following corrections for multiple testing burden. Regression models comprising several SNPs per relevant candidate gene as exploratory variables further exhibited no prediction of outcome. Our findings corroborate conclusions from previous literature about the limitations of candidate gene approaches, even when based on firm biological hypotheses, in the domain of genetic research for neuropsychiatric disorders.

showcase/). Approval to access this data is subject to a successful registration and application process to the UK Biobank. Access to the underlying dataset requires a party to set up a material transfer agreement with UK Biobank.

**Funding:** Wong, Win Lee Edwin is supported by the National University of Singapore President's Graduate Fellowship. Gavin S. Dawe's research is supported by the Ministry of Education, Singapore, under its Academic Research Fund Tier 3 Award (MOE2017-T3-1-002), and by the National Medical Research Council, Singapore, under its NMRC NUHS Centre Grant (NMRC/CG/M009/2017_NUH/ NUHS). Allan H. Young's and Cathryn Lewis' independent research is funded by the National Institute for Health and Care Research (NIHR) Maudsley Biomedical Research Centre at South London and Maudsley NHS Foundation Trust and King's College London. Ryan Arathmios' research was part funded by the National Institute for Health Research (NIHR) Maudsley Biomedical Research Centre at South London and Maudsley NHS Foundation Trust and King's College London. The funders had no role in study design, data collection and analysis, decision to publish, or preparation of the manuscript.

**Competing interests:** Gavin S. Dawe is employed by the National University of Singapore and is a co-inventor on patent applications on relaxin-3 B chain stapled peptide agonist and antagonists at RXFP3: "Stapled Peptide Agonists and Their Use in Treatment of Behavioural Disorders" Singapore Patent Application No. 10201709379P filed on 14 November 2017; and "Stapled relaxin-3 B chain peptide antagonists" Singapore Patent Application No. 10201904291Y filed on 13 May 2019. Allan H. Young declares the following competing interests. Employed by King's College London; Honorary Consultant SLaM (NHS UK). Paid lectures and advisory boards for the following companies with drugs used in affective and related disorders: AstraZeneca, Eli Lilly, Lundbeck, Sunovion, Servier, Livanova, Janssen, Allegan, Bionomics, Sumitomo Dainippon Pharma, COMPASS. Consultant to Johnson & Johnson. Consultant to Livanova. Received honoraria for attending advisory boards and presenting talks at meetings organised by LivaNova. Principal Investigator in the Restore-Life VNS registry study funded by LivaNova. Principal Investigator on ESKETINTRD3004: "An Open-label, Long-term, Safety and Efficacy Study of Intranasal Esketamine in Treatment-resistant Depression". Principal Investigator on "The Effects of Psilocybin on Cognitive Function in Healthy Participants". Principal Investigator on "The Safety and Efficacy of Psilocybin in Participants with Treatment-

## Introduction

Genetics plays a major role in neuropsychiatric conditions, with twin-based heritability estimates of major depressive disorder (MDD) between 40%-50% [1,2] and between 30%-40% [3] for anxiety disorders. Recent single nucleotide polymorphism (SNP)-based heritability measures further corroborate a role for genetics, with MDD estimates ranging between 0.102–0.162 [4]; however, the full extent of genetic involvement remains poorly defined. Much of the initial focus in this domain was on candidate gene studies, which suggested that variation in several genes was putatively implicated in MDD and anxiety disorders, like the short allele of serotonin transporter gene-linked polymorphic region [5] and Catechol-O-methyltransferase [6]. However, over the last decade, advances in computing power and sequencing technology, together with substantially larger study sample sizes, have enabled a shift towards hypothesis-free genome-wide association studies (GWAS), which have identified novel loci associated with MDD [7–9] and anxiety disorders [10–12], further underscoring the contribution of genetics to neuropsychiatric disorders. While GWAS have become the standard approach in this field, there may still be scope for candidate gene approaches, particularly for studies rooted in comprehensive understanding of biological processes.

To this end, neuropeptidergic modulation has emerged as a key contributory factor to the aetiology of several mental health disorders; for instance, much evidence implicates corticotropin-releasing factor and Substance P in affective disorders and stress-signaling [13–15]. Relaxin-3 is another such neuropeptide, identified in GABAergic neurons of the rodent nucleus incertus approximately two decades ago [16,17]. This protein binds to several receptors, including its cognate receptor RXFP3, RXFP4, and to some extent RXFP1, forming a signalling network that anatomical studies suggest is distributed across the hypothalamus, hippocampus, amygdala and several other brain regions implicated in neuropsychiatric behavioural alterations [18,19]. Moreover, several literature reviews of animal studies have highlighted the role of this relaxin-3/RXFP3 arousal system in regulating behaviours akin to neuropsychiatric conditions and their endophenotypes [19–23]; pharmacological and genetic interventions in rodent models have resulted in altered performance on several testing paradigms, as well as induced orexigenic and arousal behaviour, underlining a modulatory role for relaxin-3 signalling.

Despite evidence for the role of relaxin-3/RXFP3 in affective disorders and associated behaviours, there is a clear lack of human studies on this neuropeptidergic system. Our recent systematic review identified only five studies [23], the majority of which performed relatively poorly in methodological assessment. Limitations notwithstanding, this review identified a retrospective candidate gene study in a cohort of antipsychotic-treated patients [24], which reported several significant associations between candidate SNPs at the RLN3, RXFP3, and RXP4 genes and metabolic phenotypes, including hypertension, dyslipidaemia, and hypercholesterolaemia. Despite these preliminary findings, no follow-up work has been conducted to explore these candidate polymorphisms in a large, more robust cohort, or for any other neuropsychiatric phenotypes linked to the relaxin-3/RXFP3 system.

The UK Biobank resource [25] offers an opportunity to build on prior work and investigate these variants in a large-scale study. Participants in the UK Biobank have been deeply phenotyped, allowing for the derivation of several outcomes retrospectively. We therefore conducted a candidate gene study in UK Biobank to explore functional SNPs, determined using *in silico* prediction tools, at genes thought to be relevant to relaxin-3 signalling, including RLN3, RXFP3, RXFP4, RXFP1, and RLN2. Our outcomes of interest in this association study include depression, atypical depression, anxiety and metabolic syndrome, each of which has previously been linked to the relaxin-3/RXFP3 across the pertinent literature of animal studies. Several

Resistant Depression (P-TRD)". UK Chief Investigator for Novartis MDD study MIJ821A12201. Grant funding (past and present): NIMH (USA); CIHR (Canada); NARSAD (USA); Stanley Medical Research Institute (USA); MRC (UK); Wellcome Trust (UK); Royal College of Physicians (Edin); BMA (UK); UBC-VGH Foundation (Canada); WEDC (Canada); CCS Depression Research Fund (Canada); MSFHR (Canada); NIHR (UK). Janssen (UK). No shareholdings in pharmaceutical companies. Cathryn Lewis sits on the Scientific Advisory Board for Myriad Neuroscience, and has received speaker fees from SYNLAB. Win Lee Edwin Wong and Ryan Arathimos have no conflicts of interest to declare. This does not alter our adherence to PLOS ONE policies on sharing data and materials.

definitions of outcomes were assessed, to ensure comprehensive evaluation of our candidate variants.

## Materials and methods

### Study population

The data used in this study was collected by the UK Biobank, a study of approximately 500,000 participants across the United Kingdom [25]. Genotyping data was available for 488,171 individuals, with DNA extracted from whole blood samples then assayed using the Affymetrix UK BiLEVE Axiom Array or the Affymetrix UK Biobank Axiom Array [26]; genotype imputation was done using IMPUTE4 with Haplotype Reference Consortium (HRC) data as the main reference panel [26]. The UK Biobank has extensive phenotypic data, including self-report data and various measures from the baseline assessment, electronic health records, and further online data collection, including a mental health questionnaire [27]. Using these data, we created several definitions of depression subtypes, anxiety, and metabolic parameters, as described below.

Prior to analysis, exclusions were applied for each phenotype. Across all depression and anxiety phenotypes, participants who reported taking antipsychotics during baseline assessment interview, or with one of several confounding illnesses (mania, bipolar, schizophrenia, or psychosis) based on ICD-10 diagnostic codes from hospital inpatient data and self-reported professional diagnosis, were excluded. Controls for depression phenotypes were further excluded if they were on antidepressant treatment or fulfilled case definition for the other depression phenotypes, while controls for anxiety phenotypes were further excluded if on anxiolytic treatment or fulfilling case definitions for the other anxiety phenotypes. For metabolic phenotypes, controls were excluded when they had prescriptions of drugs corresponding to any metabolic condition comprising the metabolic syndrome definition (for example statins for hypertension).

With respect to the genetic quality control, individuals were excluded from this study when they were outliers in the genetic data for heterozygosity, had a variant call rate of <98%, had a mismatch between their reported sex and genetic sex, or were not recorded as White British on initial assessment centre visit. Moreover, individuals were excluded based on genetic relatedness, whereby only one member in groups of related individuals was used in this study; this was achieved by initial exclusion of individuals using kinship coefficients derived with the KING software, then subsequent addition of one member per related group, selecting individuals with a genetic relatedness of <0.025 with any other participant.

This study was conducted under the UK Biobank application number 16577.

### Candidate SNP selection

The process used to select candidate SNPs is outlined in **Fig 1**. The National Center for Biotechnology Information Single Nucleotide Polymorphism Database [28] was searched for all unmerged SNPs in the genes RLN3, RXFP3, RXFP4, RLN2, and RXFP1 with a minor allele frequency of >0.01, using Entrez [29] to interrogate SNP records mapped to these genes. All SNPs were then passed through the Combined Annotation-Dependent Depletion (CADD) and Genome Wide Annotation of Variants (GWAVA) *in silico* tools for identifying variants with potentially functional effects, retaining SNPs above the recommended C-score cut-off of >10 for CADD [30] and above the region, TSS, and unmatched scores cut-off of >0.5 for GWAVA [31]. Finally, the remaining SNPs were assessed for linkage disequilibrium using the LDlink web tool [32], with an $r^2$ cut-off of >0.8 used to prune SNPs in strong linkage disequilibrium based on European reference population data from the 1000 Genomes Project [33].

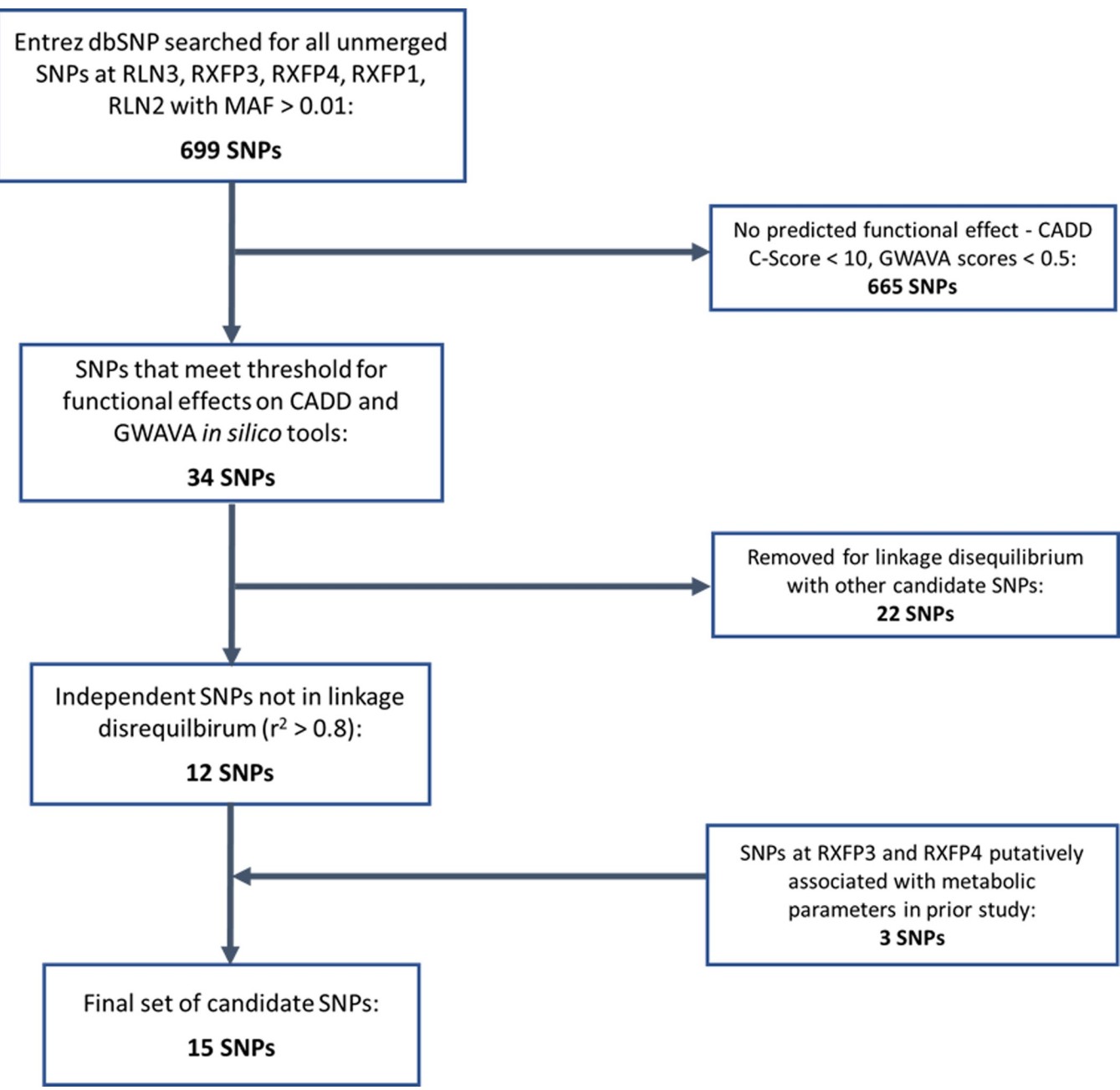

**Fig 1. Flowchart outlining the process used to select candidate single nucleotide polymorphisms in this study.**

This culminated in a final set of 12 SNPs across RLN3 (rs74400983, rs1982632, rs78161395, rs6511905), RXFP3 (rs9292519, rs171631), RXFP1 (rs62351166, rs7695640, rs11100192), and RLN2 (rs72703633, rs11793069, rs72499174). We supplemented this with an additional 3 SNPs from RXFP3 (rs42868, rs7702361) and RXFP4 (rs11264422), which were reported to be associated with metabolic parameters in a previous case-control study [24].

## Phenotype definitions

Several phenotypes were defined for the outcomes of depression, atypical depression, anxiety, metabolic syndrome, and the distinct metabolic conditions that comprise metabolic syndrome.

Depression was defined with six different phenotypes: a "broad" phenotype, an "ICD10--coded" phenotype, a "lifetime" phenotype, a CIDI phenotype, a "PHQ-9 definition" phenotype and a "PHQ-9 cut-off" phenotype. The "broad" phenotype, "ICD10-coded" phenotype and "lifetime" phenotype roughly correspond to definitions previously used in a UK Biobank GWAS of depression [8]. In brief, the broad phenotype was defined using self-reported help-seeking behaviour from participant touchscreen responses, while ICD10-coded phenotype was defined using linked hospital admission records from the UK Biobank. The lifetime phenotype was defined using a returned field from Smith et al. [34], which was derived using touchscreen responses for both help-seeking behaviour and the presence of low mood or anhedonia. The CIDI phenotype of depression [27] was derived using participant responses to the online follow-up mental health questionnaire, based on depression-relevant questions from the World Health Organisation's (WHO) Composite International Diagnostic Interview Short Form (CIDI-SF) [35], modified to consider lifetime history, where participants asked about their worst ever depression episode had to report either anhedonia or depressed mood in their lifetime and meet at least four of eight lifetime depression symptoms for caseness. The two PHQ-9 phenotypes were also derived using responses to the online follow-up mental health questionnaire, from the Patient Health Questionnaire 9-question version for current depression symptoms [36]–the "PHQ-9 definition" was based on a major depression diagnosis on the PHQ-9, while the "PHQ-9 cut-off" phenotype was based on a cut-off score of $\geq$10 [36,37].

The atypical depression subtype was defined with three different phenotypes, for participants who responded to the online follow-up mental health questionnaire. Caseness for atypical depression was defined as an indication of depression, based on our previously defined CIDI phenotype, PHQ-9 definition phenotype, or PHQ-9 cut-off definition phenotype, as well as indicating both hypersomnia and weight gain during their worst episode of depression on the online follow-up mental health questionnaire [38]. Anxiety was defined with three different phenotypes: an ICD10-coded phenotype, a "lifetime disorder" phenotype and a "GAD-7 cut-off" phenotype, the latter two of which mirror the definitions used in a previous GWAS of anxiety [12]. The ICD10-coded phenotype of anxiety was defined using linked hospital admission records for the main five anxiety disorders. The lifetime disorder phenotype was derived with a combination of self-reported lifetime professional diagnosis for the main five anxiety disorders and responses to the online follow-up mental health questionnaire, based on anxiety-relevant questions from the CIDI. The GAD-7 cut-off phenotype was defined using responses to answers to questions based on the Generalised Anxiety Disorder Assessment (GAD-7), using a cut-off score of $\geq$10 to define caseness [39].

Metabolic syndrome was assessed using the National Heart, Lung and Blood Institute definition [40], where caseness was defined as having $\geq$3 of the following risk factors: a waist circumference of $\geq$102 cm for males or $\geq$88 cm for females, hypertension (or antihypertensive drug treatment), hypertriglyceridaemia (or drug treatment for elevated triglycerides), hyperglycaemia (or drug treatment for elevated glucose) and low HDL cholesterol levels (or drug treatment for reduced HDL cholesterol). Each of these risk factors, with the exception of waist circumference, were also assessed as individual phenotypes.

The full criteria used to derive these phenotype variables in the UK Biobank, including all relevant field codes used, are detailed in **S1**–**S3 Tables**.

## Statistical analyses

We performed logistic regression to test for association between each candidate gene and outcome. These regression models were adjusted for age, age$^2$, sex, genotyping batch, testing centre, and the first six European ancestry principal components (to control for population structure). For each primary outcome of interest (i.e., MDD, anxiety, atypical depression and metabolic syndrome), we corrected for multiple testing using false discovery rate (FDR) across the various phenotype definitions– 6 definitions for MDD, 3 definitions for anxiery, 3 definitions for atypical depression and 5 definitions for metabolic syndrome–and compared these q-values against a threshold of 0.01, more stringent than the standard threshold of 0.05 to account for the multiple SNPs tested. To ensure that there were no methodological errors in phenotypic definitions or exclusions, similar regression testing was also conducted for three independent, control SNPs at chromosome 7 from a previous GWAS analysis done for MDD in the UK Biobank: rs3807865, rs1554505, and rs5011432. All three of these polymorphisms were previously significantly associated with a specific MDD outcome phenotype, such that these are solid control points of reference to verify that the correct quality control procedures were implemented in the present study.

We also conducted a sensitivity analysis for each candidate gene, in which every candidate SNP at a particular gene (for RLN3, RXFP3, RXFP1, and RLN2) was an explanatory variable in a single multivariable regression model, adjusted for covariates as described above. This was then compared to an analogous regression model with no candidate SNP variables present, using a likelihood ratio test. This was conducted for all outcome phenotypes, with multiple testing accounted for as in the individual SNP analyses, with an FDR correction across the number of phenotype sub-definitions then comparison to an alpha level of 0.01.

# Results

## Study demographics

In total, there were six phenotypes for depression, three phenotypes for anxiety, three phenotypes for atypical depression, and five phenotypes for metabolic indications. Study demographics for case and control groups in each phenotype definition are outlined in **Table 1**. The number of cases and controls within each phenotype varied heavily, as each was derived using a different set of responses in UK Biobank. Across all phenotype definitions, the mean age of participants at baseline ranged from 51.6 years to 59.2 years.

## SNP associations with depression and anxiety phenotypes

Each candidate SNP was individually investigated for all phenotypes, across all outcomes of interest. When corrected for the number of tests conducted, there were no associations between any individual candidate SNP and any of the six depression phenotypes (**Fig 2**). Before correction for multiple testing, rs74400983 (unadjusted p = 0.0232) and rs78161395 (unadjusted p = 0.0428) at RLN3, as well as rs62351166 (unadjusted p = 0.0388) at RXFP1 were nominally linked with the broad phenotype of depression; however, these did not withstand FDR corrections. No candidate SNPs were associated with any of the three atypical depression phenotypes following corrections for multiple test burden (**Fig 3**), with rs42868 from RXFP3 closest to demonstrating any significant association (unadjusted p = 0.0297).

In addition to evaluating these candidate SNPs, we also tested the SNPs rs3807865, rs1554505, and rs5011432 at chromosome 7, which have previously been associated with single depression phenotype definitions (that mirror our definitions) in a 2018 GWAS of MDD in the UK Biobank [8]. The associations from this GWAS were largely reproduced in this study,

**Table 1. Summary of study demographics for case and control groups in each of the phenotypic definitions, across all outcomes.**

| Phenotype | Total | Number of cases | Number of controls | %Male | Mean age (standard deviation) |
|---|---|---|---|---|---|
| Broad depression | 367445 | 131219 | 236226 | 53.9% | 56.8 (8.01) |
| ICD10-coded depression | 288937 | 19307 | 269630 | 53.0% | 57.3 (7.98) |
| Lifetime depression | 85881 | 23271 | 62610 | 52.8% | 57.2 (8.01) |
| CIDI depression | 111786 | 33449 | 78337 | 56.1% | 56.1 (7.67) |
| PHQ-9 definition depression | 103236 | 5470 | 97766 | 54.5% | 56.2 (7.69) |
| PHQ-9 cutoff depression | 105170 | 5720 | 99450 | 54.9% | 56.1 (7.7) |
| CIDI atypical depression | 79999 | 1662 | 78337 | 51.3% | 56.6 (7.67) |
| PHQ-9 definition atypical depression | 98192 | 426 | 97766 | 54.4% | 56.3 (7.66) |
| PHQ-9 cutoff atypical depression | 100011 | 561 | 99450 | 54.4% | 56.3 (7.67) |
| ICD10-coded anxiety | 321083 | 16151 | 304932 | 54.3% | 57.2 (7.97) |
| Lifetime disorder anxiety | 101836 | 21045 | 80791 | 56.1% | 56.2 (7.68) |
| GAD-7 cutoff anxiety | 114667 | 4472 | 110195 | 56.1% | 56.1 (7.67) |
| Metabolic syndrome | 305742 | 104629 | 201113 | 53.4% | 56.7 (8.02) |
| Hypertension | 335730 | 235005 | 100725 | 54.0% | 56.6 (8.05) |
| Low HDL cholesterol | 336469 | 65720 | 270749 | 53.5% | 56.7 (8.02) |
| Hyperglycaemia | 331715 | 25573 | 306142 | 53.7% | 56.7 (8.03) |
| Hypertriglyceridaemia | 326585 | 147594 | 178991 | 54.7% | 56.2 (8.05) |

with rs3807865 still significantly associated with the broad phenotype depression ($\beta$ = 0.033, SE = 0.005, p = $4.74 \times 10^{-11}$), rs5011432 significantly associated with the lifetime phenotype depression ($\beta$ = 0.048, SE = 0.011, p = $1.96 \times 10^{-5}$), and rs1554505 significantly associated with the ICD10-coded depression phenotype ($\beta$ = -0.047, SE = 0.013, p = $2.04 \times 10^{-4}$).

There were also no associations between any individual candidate SNP and any of the three anxiety phenotypes (Fig 4). Before applying our relatively liberal multiple test correction procedure, rs11264422 from RXFP4 was preliminarily associated with both our probable (unadjusted p = 0.00318) and ICD10-coded (unadjusted p = 0.0106) phenotype for anxiety; however, following FDR corrections, these associations were not significant against our threshold of 0.01.

Detailed information for all relevant regression analyses pertinent to depression, atypical depression, and anxiety phenotypes are summarised in S4–S6 Tables.

## SNP associations with metabolic phenotypes

Candidate SNPs were also tested for associations with metabolic syndrome and four of its constituent phenotypes: hypertension, low HDL cholesterol levels (dyslipidaemia), hypertriglyceridaemia, and hyperglycaemia. After FDR adjustment for the number of sub-phenotypes tested, there were no associations between any of the candidate SNPs and any of the metabolic phenotypes (Fig 5). Associations between rs72499174 from RLN2 and two phenotypes–hypertension (q = 0.0355) and hypertriglyceridaemia (q = 0.0704)–were the two closest to the significance cut-offs used. The candidate SNPs rs42868 (unadjusted p = 0.0289) and rs11793069 (unadjusted p = 0.0353) were also provisionally linked with low HDL cholesterol levels (dyslipidaemia) and hypertension, respectively. However, these associations were not statistically significant after corrections for multiple testing. Detailed information for all pertinent regression analyses involving metabolic phenotypes is summarised in S7 Table.

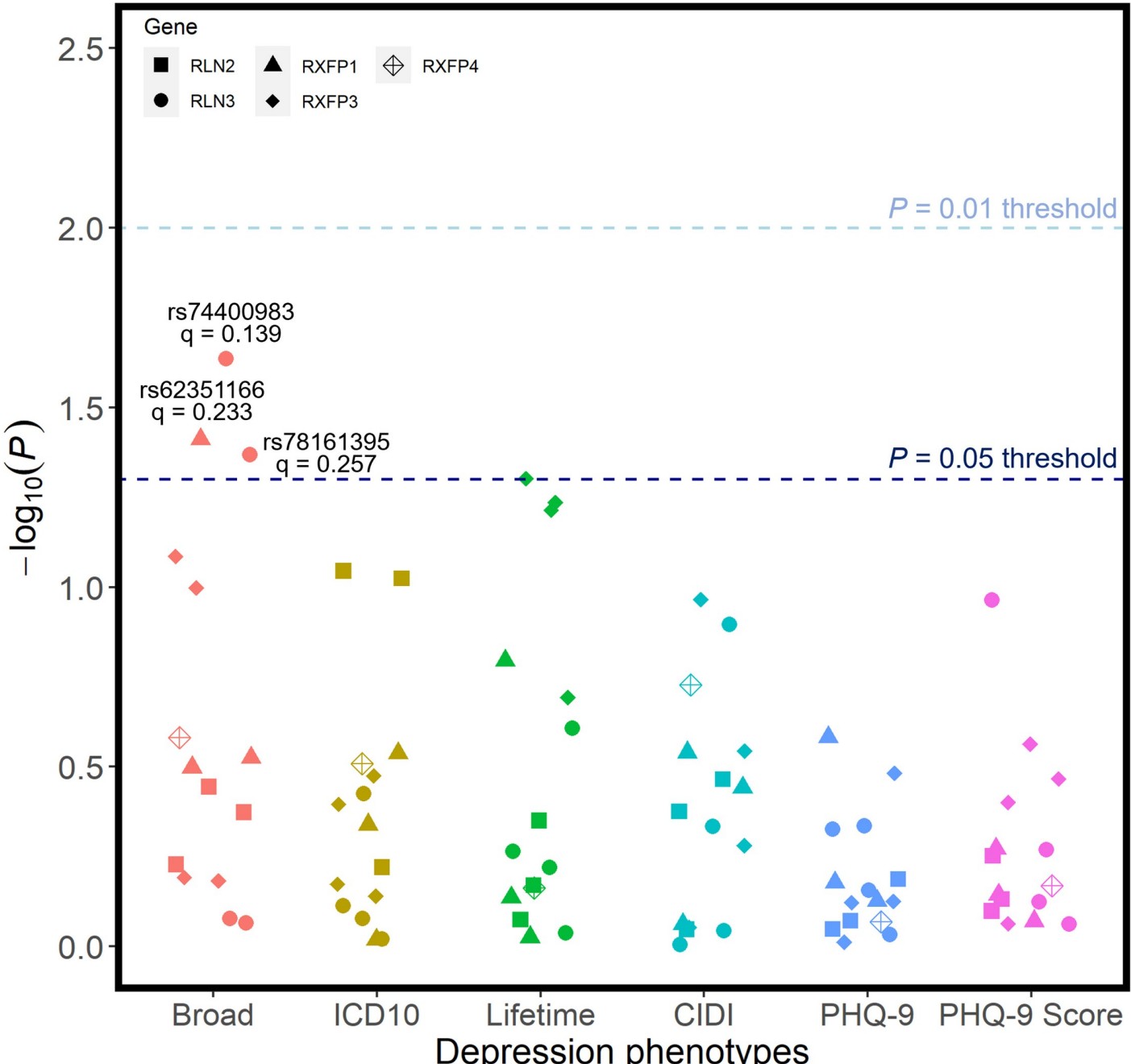

**Fig 2. Unadjusted p-values for association tests between all SNPs and the 6 phenotypic definitions of depression.** Regression models were adjusted for age, age[2], sex, genotyping batch, testing centre, and the first six European ancestry principal components. SNPs that were nominally associated with a phenotype are annotated with their corresponding q-value following FDR correction–there were no statistically significant associations.

### Sensitivity analyses by gene

Association analyses for each gene were also conducted to additionally investigate our candidate SNPs of interest. All SNPs at a particular gene were entered as explanatory variables in a single multivariate logistic regression model, for each outcome phenotype, which was then compared to a null model with no SNPs and only covariates as predictors. Combined SNPs at

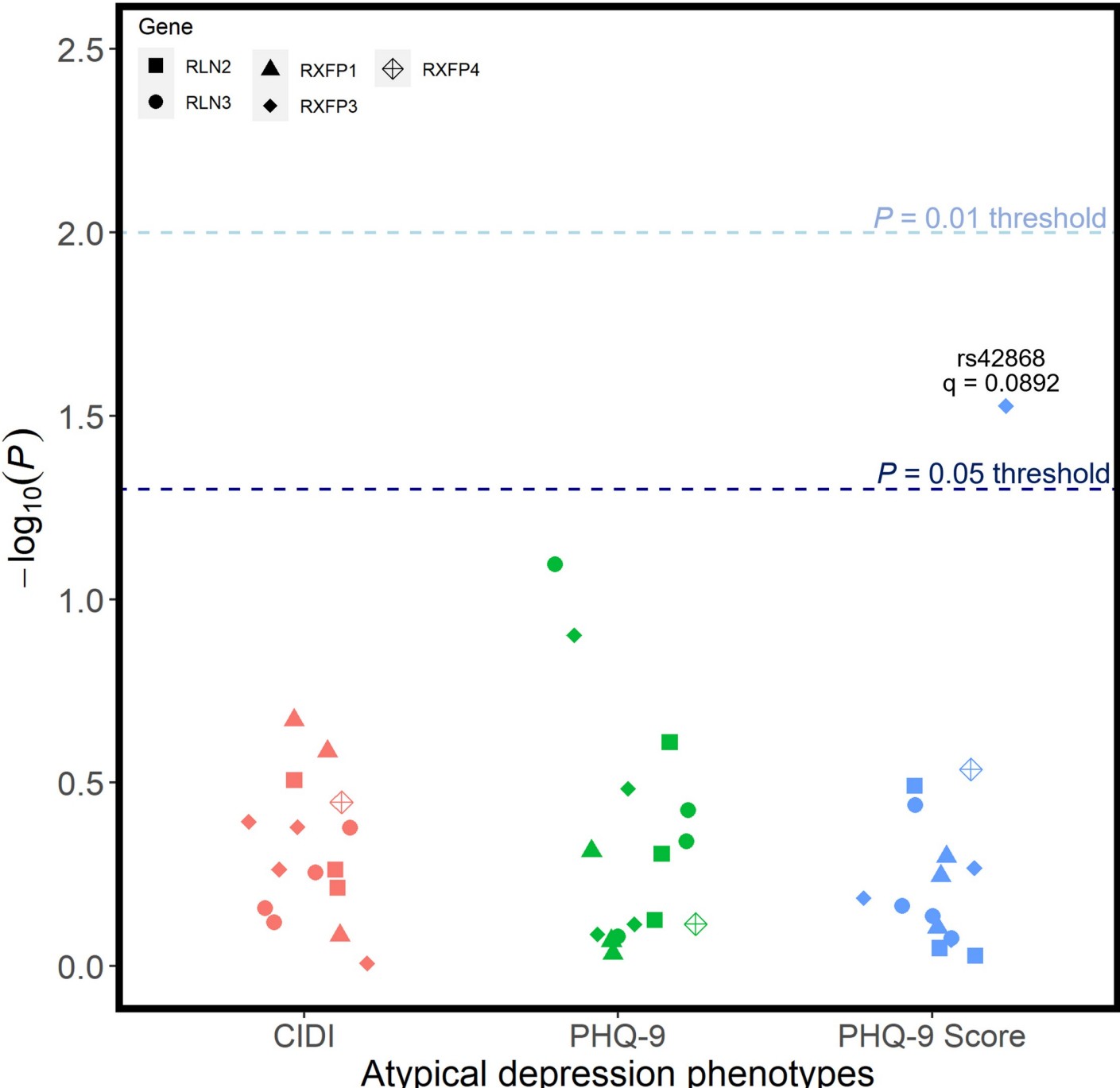

**Fig 3. Unadjusted p-values for association tests between all SNPs and the 3 phenotypic definitions of atypical depression.** Regression models were adjusted for age, age2, sex, genotyping batch, testing centre, and the first six European ancestry principal components. SNPs that were nominally associated with a phenotype are annotated with their corresponding q-value following FDR correction–there were no statistically significant associations.

RLN3, RXFP3, RLN2, and RXFP1 were tested, with RXFP4 not included as there was only one pertinent candidate SNP at this gene. Across all depression, anxiety, atypical depression and metabolic phenotypes, none of the regression models comprising multiple SNP predictors significantly improved fit on the likelihood ratio test, compared to the null models, following

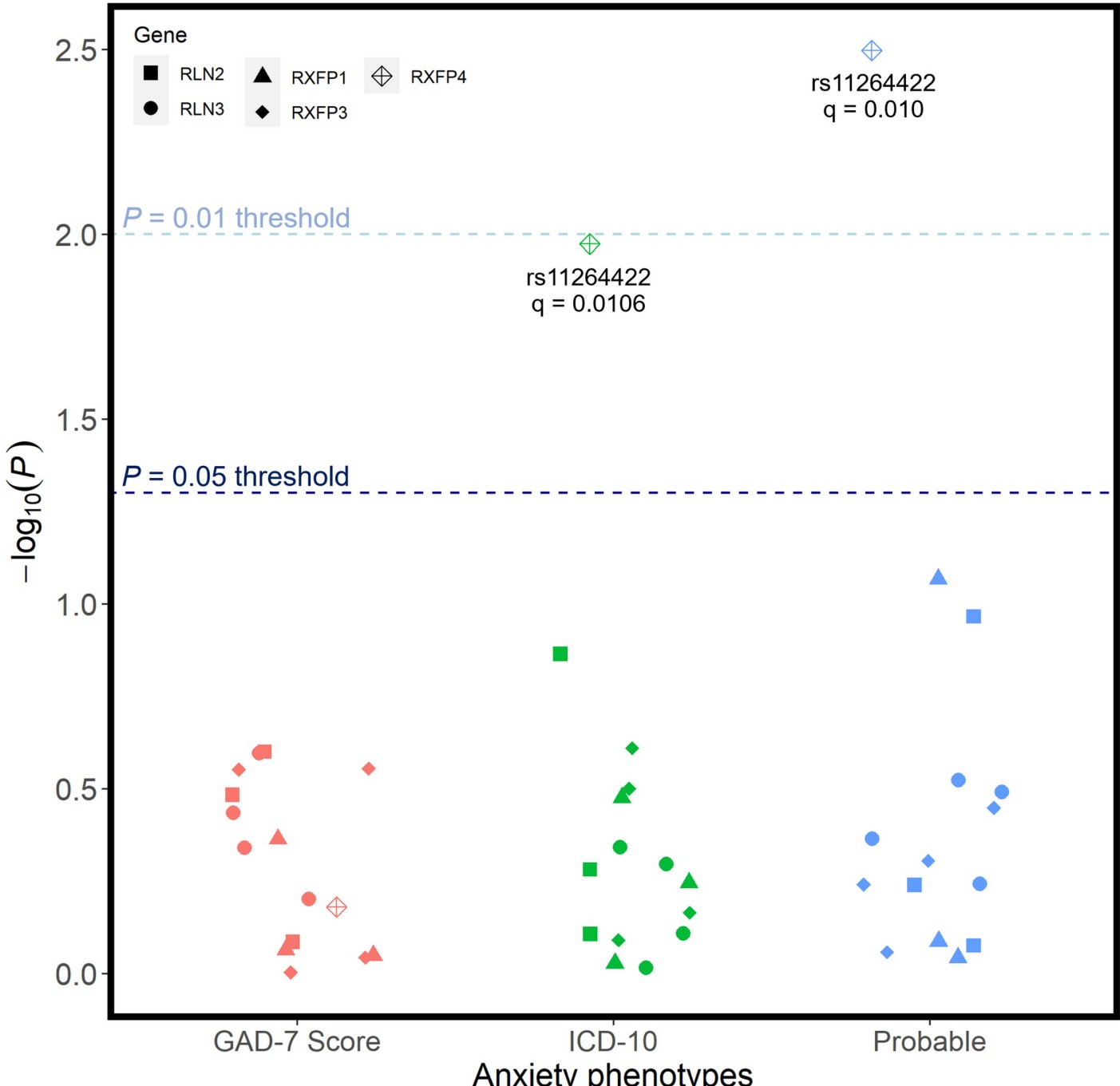

**Fig 4. Unadjusted p-values for association tests between all SNPs and the 3 phenotypic definitions of anxiety disorders.** Regression models were adjusted for age, age2, sex, genotyping batch, testing centre, and the first six European ancestry principal components. SNPs that were nominally associated with a phenotype are annotated with their corresponding q-value following FDR correction–there were no statistically significant associations.

correction for multiple testing burden (**S8**–**S10 Tables**). This further underscores the lack of observed associations between our candidate SNPs and the outcomes of interest in the UK Biobank.

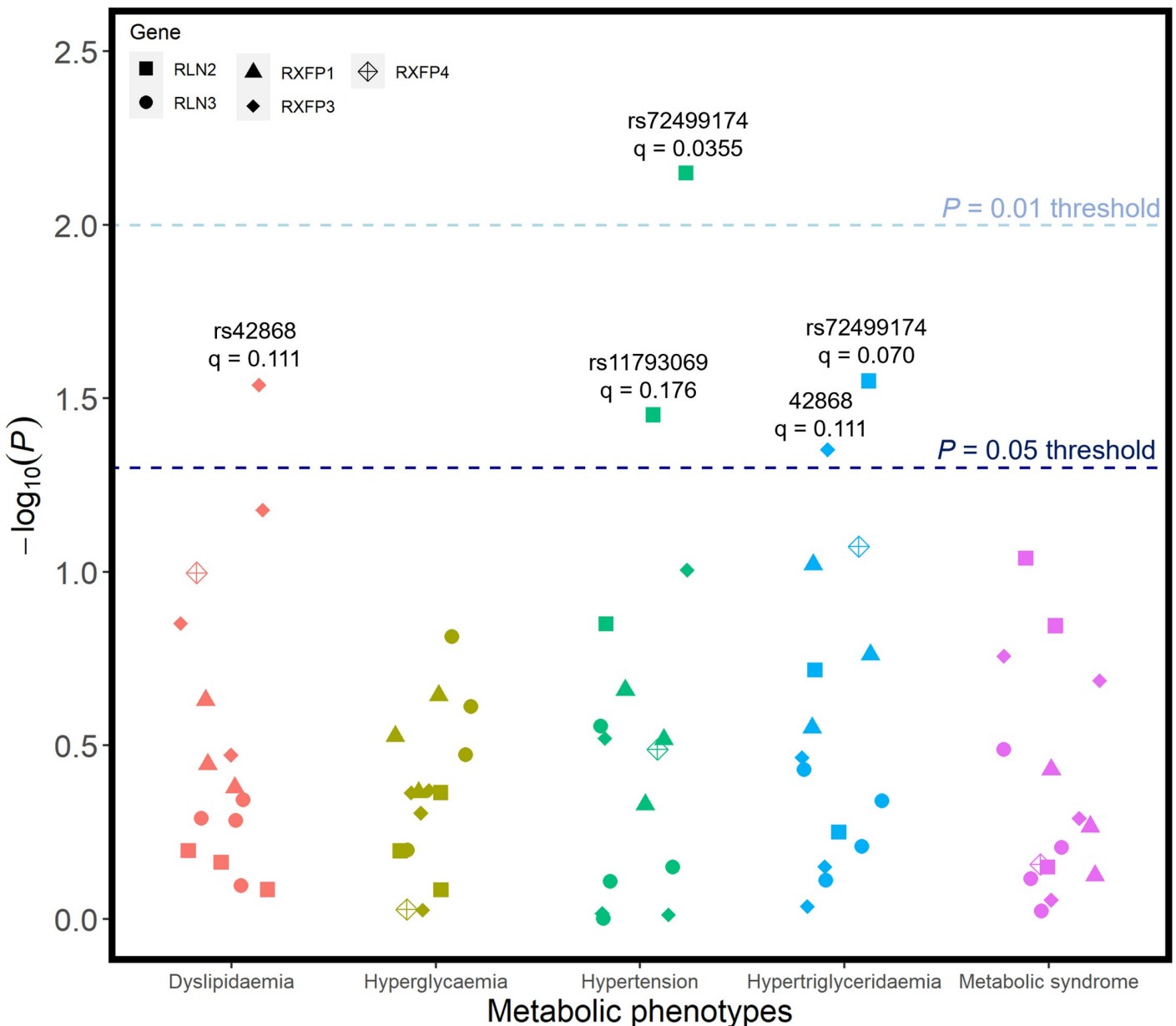

**Fig 5. Unadjusted p-values for association tests between all SNPs and metabolic syndrome, as well as 4 sub-outcomes that comprise metabolic syndrome.** Regression models were adjusted for age, age2, sex, genotyping batch, testing centre, and the first six European ancestry principal components. SNPs that were nominally associated with a phenotype are annotated with their corresponding q-value following FDR correction–there were no statistically significant associations.

## Discussion

In this analysis of a large population-based cohort, a set of functional SNPs from several candidate genes relevant to relaxin-3 signalling were investigated for associations with depression, anxiety, atypical depression, and metabolic syndrome, corresponding to a putative link reported in the animal literature. However, across all outcomes, there were no significant associations between any individual candidate SNP and any of the outcome phenotypes, suggesting that common genetic variation in the relaxin-3/RXFP3 system makes little contribution to

liability for any of our outcomes of interest. This lack of associations arose in spite of the relatively lenient procedure undertaken to correct for multiple testing, in which an FDR correction was applied across the several phenotype definitions per outcome and compared against a threshold of 0.01, without detailed consideration for the number of SNPs tested. An FDR approach was adopted to enable the detection of any possible signals in this study, yet no significant associations were identified. Furthermore, the present results cannot be attributed to errors in the data preparation and analysis pipeline undertaken in this study, as previously identified GWAS hits in the UK Biobank were reproduced in a sensitivity analysis. A lack of statistical power cannot be ruled out as a potential factor explaining the absence of any significant associations, particularly for certain phenotypes, like the PHQ-9 cut-off phenotype of depression, for which there were only 5,720 cases; however, sample sizes across the different phenotypic subsamples ranged between 85,881 and 386,769, rendering this analysis relatively well powered, especially when compared to the traditional candidate gene literature in this domain. Moreover, we used between three to six phenotypic definitions under each umbrella outcome, a very comprehensive approach to ensure that any findings were not specific to a restricted outcome sub-definition; for instance, the six phenotypes of depression help to circumvent some of the diagnostic heterogeneity [41,42] of the disorder. The consistent lack of associations across the different phenotypes for each definition, along with the reasonably high power of this UK Biobank analysis, further reinforce the notion that our results are attributable to the absence of any direct link between genetic variations at the relaxin-3/RXFP3 system and the presence of neuropsychiatric and metabolic outcomes.

These findings conflict with the only prior candidate gene study in the relaxin-3/RXFP3 field, which reported associations between several candidate SNPs and miscellaneous metabolic parameters [24]. The only variant peripherally pertinent in both studies was rs42868 at RXFP3, which was associated with diabetes and hypercholesterolaemia in the prior study and nominally associated with low HDL cholesterol levels (dyslipidaemia) in this analysis. There are several potential explanations for the discrepancies between these two studies. Firstly, the previous study was conducted using a cohort of 419 antipsychotic-treated patients, a niche sample with fewer participants than our present investigation of the UK Biobank. Secondly, the metabolic outcome definitions differed between studies: the prior study used BMI, clinical records of diabetes and hypercholesterolaemia, as opposed to the present study, which used waist circumference, fasting glucose levels, and HDL cholesterol levels; our definitions were derived based on definitions previously used for other analyses of metabolic syndrome in the UK Biobank [43,44], and are in line with the standardised definition from the National Heart, Lung, and Blood Institute [40]. Thirdly, the candidate polymorphisms investigated in the two studies did not overlap completely: the prior study made use of the HapMap programme [45], whereas we leveraged several *in silico* prediction tools that were developed only after the initial 2011 study, and explored the additional relaxin family genes RLN2 and RXFP1 [46]. Nonetheless, we supplemented our set of SNPs with the prior study's putatively reported significant candidate SNPs, to ensure our investigation encompassed all prior reported associations. Perhaps the biggest contributor to the difference in results was the statistical methodology adopted. While both studies employed multivariable regressions, there were differences in the external variables controlled for; our study controlled for potential genotyping differences, as well as population structure (using principal components), but did not take into account duration of antipsychotic medication, which was pertinent in the prior study given the nature of its cohort. Furthermore, our study corrected for multiple testing burden using FDR, whereas the prior study did not use any multiple test corrections. It is important to note that had the prior study corrected for multiple testing, using for instance a lenient Bonferroni approach across phenotypes or SNPs, no SNP would have been significantly associated. While there is no clear

consensus on the appropriate approaches to correcting for numerous tests [47,48], a complete lack of correction is not robust considering the number of polymorphisms and outcomes tested. As such, the prior findings are likely to have been false positives generated by chance, and our present results are more reflective of the relationship (or lack thereof) between genetic variation relevant to relaxin-3 signalling and metabolic outcomes.

More broadly, these results provide additional support for a shift away from candidate gene studies in psychiatric research. An extensive and well-powered investigation from Border et al. [49] evaluated 18 commonly studied candidate genes from the last three decades for associations with several depressive phenotypes, finding that there was little evidence of any relationship between the candidate polymorphisms and depression liability. This mirrors conclusions from other rigorous studies and reviews; for example, van de Weijer et al. [50] highlighted a lack of support for candidate gene associations with well-being, a construct heavily related to neuropsychiatric disorders. Our focused candidate gene study on the relaxin-3 neuropeptidergic system corroborate these conclusions, underlining the importance of proceeding cautiously before undertaking a candidate gene approach. Genetic contributions towards multifactorial disorders, like depression, anxiety, or metabolic syndrome, are extremely complex and polygenic in nature, with our study further demonstrating the minuscule effect sizes that individual genes have on these phenotypes. While candidate gene approaches may still be relevant in particular scenarios, our findings underscore why there has been a shift towards GWAS approaches in understanding the genetic underpinnings of complex diseases.

It is important to recognise that a lack of significant associations in this candidate gene study does not rule out a role for the relaxin-3/RXFP3 system in MDD, atypical depression, anxiety or metabolic syndrome. The Border et al. study [49] explored several key neurotransmitters and peptides that are generally accepted to be involved in the aetiology of MDD, yet also revealed a lack of associations between polymorphisms at the pertinent genes and presence of MDD. For example, SNPs from the HTR2A gene, which codes for the serotonin 5-HT2A receptor, were not significantly associated with MDD. This finding does not, however, invalidate prior evidence outlining 5-HT2A receptor involvement in MDD, with good evidence for the distribution of this protein across brain areas related to MDD and several preclinical studies linking this receptor with depressive-like phenotypes [51,52]. The apparent inconsistency between null genetic associations and putative pathophysiological involvement is not altogether surprising as these complex, polygenic diseases are the culmination of dynamic interactions between several peptidergic systems, such that variation at only a small subset of genes may not be particularly informative, highlighted by the small effect sizes. Moreover, these candidate gene approaches only explore genetic differences at the DNA level, and do not consider epigenetic factors, which could very feasibly, in this context, alter the expression levels of the various relaxin-3/RXFP3-relevant proteins.

The present investigation possesses several strengths. This is the first study to evaluate candidate SNPs pertinent to the relaxin-3/RXFP3 system in the context of depression, anxiety and atypical depression, and the first to assess a relationship with metabolic parameters using a rigorous statistical approach. The relatively large sample size for the analyses is a particular strength relative to the traditional candidate gene literature in this domain. Furthermore, we were cognisant to define several phenotypes for each of our outcomes of interest; the consistency in our null results across the various phenotypic definitions provides some assurance that our findings are reflective of a holistic definition for MDD, atypical depression, anxiety, and metabolic syndrome.

There were also several important limitations of note in this study. Firstly, our candidate gene selection methodology is rooted in the use of *in silico* tools, which have certain limitations in predicting variant deleteriousness. We employed two different tools, each based on different

statistical methodology and principles, to help overcome the lack of absolute certainty in predicting the effects of variants. The cut-offs used in this study were recommended by the creators of each tool and helped ensure we attained a reasonable number of candidate SNPs to evaluate. There are also several limitations to our outcome phenotype definitions, though the use of proxies to define our disease states of interest was unavoidable. Several of the phenotype definitions for depression and anxiety are based on participant self-reported information, which can be prone to biases that may arise from elements of recall and social desirability [53]. The broad definition of depression in particular may additionally capture an overly general phenotype, given the frequency of comorbidity between depression and anxiety disorders. Our CIDI phenotype of definition also did not stringently mirror the diagnostic criteria from the WHO CIDI assessment, as one of the nine pertinent depression symptoms (psychomotor agitation or retardation) was not available in the online mental health follow-up questionnaire. Consequently, our CIDI phenotype altered the standard definition of $\geq 5$ of 9 symptoms to $\geq 4$ of 8 symptoms. While our definitions for atypical depression have been used in previous analyses of the UK Biobank [38], they are also of ambiguous reliability, as they were based on answers to questions about hypersomnia and weight gain, which do not comprise the full spectrum of atypical symptoms that characterise this disorder [54–56]. The limitations inherent in each of these definitions provides further rationale for the wide range of outcome phenotype definitions used in this analysis, including definitions that mirror the validated PHQ-9 and GAD-7 questionnaires [36,39]. Finally, it is important to acknowledge that the UK Biobank is not a robust representation of the general UK population, with higher participation rates across certain demographics [57]; this may limit the generalisability of these findings.

In summary, this candidate gene study revealed that candidate functional polymorphisms at RLN3, RXFP3, RXFP4, RLN2, and RXFP1 had no significant effects on the outcomes of MDD, atypical depression, anxiety, and metabolic syndrome. This lack of associations was consistent across the many phenotypic definitions investigated, ensuring comprehensive analysis, and further confirmed in sensitivity analyses exploring several candidate polymorphisms simultaneously. While the relaxin-3/RXFP3 system may still be involved in the pathophysiology of these diseases, this is unlikely to be reflected in common genetic variation at the DNA level. At a broader level, our findings support prior conclusions for prudent consideration and intereptation of candidate gene studies for neuropsychiatric or metabolic conditions, especially given their complex underpinnings,.

## Supporting information

**S1 Table. Full description and field codes used to derive phenotype caseness definitions for depression and atypical depression.**
(DOCX)

**S2 Table. Full description and field codes used to derive phenotype caseness definitions for anxiety.**
(DOCX)

**S3 Table. Full description and field codes used to derive phenotype caseness definitions for metabolic syndrome and the sub-outcomes that comprise the disorder.**
(DOCX)

**S4 Table. Full associations between each candidate SNP and each of the 6 phenotypic definitions for depression.** Regression models were adjusted for age, age$^2$, sex, genotyping batch, testing centre, and the first six European ancestry principal components. Unadjusted p values and q-values (calculated by applying false discovery rate correction across phenotype

definitions) are presented.
(DOCX)

**S5 Table. Full associations between each candidate SNP and each of the 3 phenotypic definitions for atypical depression.** Regression models were adjusted for age, age$^2$, sex, genotyping batch, testing centre, and the first six European ancestry principal components. Unadjusted p values and q-values (calculated by applying false discovery rate correction across phenotype definitions) are presented.
(DOCX)

**S6 Table. Full associations between each candidate SNP and each of the 3 phenotypic definitions for anxiety disorders.** Regression models were adjusted for age, age$^2$, sex, genotyping batch, testing centre, and the first six European ancestry principal components. Unadjusted p values and q-values (calculated by applying false discovery rate correction across phenotype definitions) are presented.
(DOCX)

**S7 Table. Full associations between each candidate SNP and metabolic syndrome, as well as 4 sub-outcomes that comprise metabolic syndrome.** Regression models were adjusted for age, age$^2$, sex, genotyping batch, testing centre, and the first six European ancestry principal components. Unadjusted p values and q-values (calculated by applying false discovery rate correction across phenotype definitions) are presented.
(DOCX)

**S8 Table. Results of a multivariate regression model with all candidate SNPs at a particular gene as simultaneous explanatory variables and several depression and atypical depression phenotype outcomes.** Models were adjusted for age, age$^2$, sex, genotyping batch, testing centre, and the first six European ancestry principal components. Unadjusted p values and q-values (calculated by applying false discovery rate correction across phenotype definitions) are presented.
(DOCX)

**S9 Table. Results of a multivariate regression model with all candidate SNPs at a particular gene as simultaneous explanatory variables and several anxiety phenotype outcomes.** Models were adjusted for age, age$^2$, sex, genotyping batch, testing centre, and the first six European ancestry principal components. Unadjusted p values and q-values (calculated by applying false discovery rate correction across phenotype definitions) are presented.
(DOCX)

**S10 Table. Results of a multivariate regression model with all candidate SNPs at a particular gene as simultaneous explanatory variables and metabolic syndrome, as well as the sub-outcomes comprising the disorder.** Models were adjusted for age, age$^2$, sex, genotyping batch, testing centre, and the first six European ancestry principal components.
(DOCX)

## Acknowledgments

The views expressed are those of the author(s) and not necessarily those of the NHS, the NIHR or the Department of Health and Social Care. This research has been conducted using the UK Biobank Resource under Application Number 16577. The authors wish to thank Jonathan R. I. Coleman and Gerome Breen for their valuable contributions in support of this study. In addition, we

are grateful to all UK Biobank staff and volunteers. Finally, the authors acknowledge use of the research computing facility at King's College London, Rosalind (https://rosalind.kcl.ac.uk).

## Author Contributions

**Conceptualization:** Win Lee Edwin Wong, Allan H. Young, Gavin S. Dawe.

**Formal analysis:** Win Lee Edwin Wong.

**Funding acquisition:** Cathryn M. Lewis, Allan H. Young, Gavin S. Dawe.

**Investigation:** Win Lee Edwin Wong, Ryan Arathimos, Cathryn M. Lewis.

**Methodology:** Win Lee Edwin Wong, Ryan Arathimos.

**Project administration:** Win Lee Edwin Wong, Ryan Arathimos, Cathryn M. Lewis.

**Resources:** Cathryn M. Lewis.

**Supervision:** Cathryn M. Lewis, Allan H. Young, Gavin S. Dawe.

**Writing – original draft:** Win Lee Edwin Wong.

**Writing – review & editing:** Win Lee Edwin Wong, Ryan Arathimos, Cathryn M. Lewis, Allan H. Young, Gavin S. Dawe.

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
