## [Decision Letter · Decision Letter 0]

25 Oct 2023

Investigating the role of the relaxin-3/RXFP3 system in neuropsychiatric disorders and metabolic phenotypes: a candidate gene approach

PONE-D-23-28052

Dear Dr. Wong,

We’re pleased to inform you that your manuscript has been judged scientifically suitable for publication and will be formally accepted for publication once it meets all outstanding technical requirements.

Kind regards,

Irina U Agoulnik, Ph.D.

Academic Editor

PLOS ONE

 [Wong, Win Lee Edwin is supported by the National University of Singapore President’s Graduate Fellowship. Gavin S. Dawe’s research is supported by the Ministry of Education, Singapore, under its Academic Research Fund Tier 3 Award (MOE2017-T3-1-002), and by the National Medical Research Council, Singapore, under its NMRC NUHS Centre Grant (NMRC/CG/M009/2017_NUH/NUHS). Allan H. Young’s and Cathryn Lewis’  independent research is funded by the National Institute for Health and Care Research (NIHR) Maudsley Biomedical Research Centre at South London and Maudsley NHS Foundation Trust and King’s College London. Ryan Arathmios’ research was  part funded by the National Institute for Health Research (NIHR) Maudsley Biomedical Research Centre at South London and Maudsley NHS Foundation Trust and King’s College London.]  

Please respond by return e-mail so that we can amend your financial disclosure and competing interests on your behalf.

[Gavin S. Dawe is employed by the National University of Singapore and is a co-inventor on patent applications on relaxin-3 B chain stapled peptide agonist and antagonists at RXFP3: “Stapled Peptide Agonists and Their Use in Treatment of Behavioural Disorders” Singapore Patent Application No. 10201709379P filed on 14 November 2017; and “Stapled relaxin-3 B chain peptide antagonists” Singapore Patent Application No. 10201904291Y filed on 13 May 2019.

Allan H. Young declares the following competing interests. Employed by King’s College London; Honorary Consultant SLaM (NHS UK). Paid lectures and advisory boards for the following companies with drugs used in affective and related disorders: AstraZeneca, Eli Lilly, Lundbeck, Sunovion, Servier, Livanova, Janssen, Allegan, Bionomics, Sumitomo Dainippon Pharma, COMPASS. Consultant to Johnson & Johnson. Consultant to Livanova. Received honoraria for attending advisory boards and presenting talks at meetings organised by LivaNova. Principal Investigator in the Restore-Life VNS registry study funded by LivaNova. Principal Investigator on ESKETINTRD3004: “An Open-label, Long-term, Safety and Efficacy Study of Intranasal Esketamine in Treatment-resistant Depression”. Principal Investigator on “The Effects of Psilocybin on Cognitive Function in Healthy Participants”. Principal Investigator on “The Safety and Efficacy of Psilocybin in Participants with Treatment-Resistant Depression (P-TRD)”. UK Chief Investigator for Novartis MDD study MIJ821A12201. Grant funding (past and present): NIMH (USA); CIHR (Canada); NARSAD (USA); Stanley Medical Research Institute (USA); MRC (UK); Wellcome Trust (UK); Royal College of Physicians (Edin); BMA (UK); UBC-VGH Foundation (Canada); WEDC (Canada); CCS Depression Research Fund (Canada); MSFHR (Canada); NIHR (UK). Janssen (UK). No shareholdings in pharmaceutical companies.

Cathryn Lewis sits on the Scientific Advisory Board for Myriad Neuroscience, and has received speaker fees from SYNLAB.  

Win Lee Edwin Wong and Ryan Arathimos have no conflicts of interest to declare.]. 

Please respond by return email with your amended Competing Interests Statement and we will change the online submission form on your behalf.

5. We noted in your submission details that a portion of your manuscript may have been presented or published elsewhere. [DETAILS AS NEEDED] Please clarify whether this [conference proceeding or publication] was peer-reviewed and formally published. If this work was previously peer-reviewed and published, in the cover letter please provide the reason that this work does not constitute dual publication and should be included in the current manuscript.

6. Please include your tables as part of your main manuscript and remove the individual files. Please note that supplementary tables (should remain/ be uploaded) as separate ""supporting information"" files".

Reviewers' comments:

Reviewer's Responses to Questions

**Comments to the Author**

1. Is the manuscript technically sound, and do the data support the conclusions?

Reviewer #1: Yes

Reviewer #2: Yes

2. Has the statistical analysis been performed appropriately and rigorously? 

Reviewer #1: Yes

Reviewer #2: Yes

3. Have the authors made all data underlying the findings in their manuscript fully available?

Reviewer #1: Yes

Reviewer #2: Yes

4. Is the manuscript presented in an intelligible fashion and written in standard English?

Reviewer #1: Yes

Reviewer #2: Yes

5. Review Comments to the Author

Reviewer #1: The manuscript by Wong et al describes an attempt to link SNPs in RLN3 gene and its receptors RXFP1/3/4 as well as RLN2 gene with depression, atypical depression, anxiety, and various features of metabolic syndrome in large population samples from UK Biobank. Previous data derived from rodent studies suggest the putative role of this relaxin peptide signaling on behavioral phenotype. However, such a candidate gene approach did not reveal any significant associations between SNPs and human conditions. Nevertheless, I believe that even with the negative results the manuscript is suitable for publication. First, clear definitions of the phenotypes were used to select affected patients. Second, the population size is large enough to detect such associations. Third, the method was controlled by applying the same approach to the Chr 7 SNPs previously linked to depression in the GWAS study. In that case, those SNPs have been again linked to the phenotype. The manuscript is well-written and is easy to follow. I find the discussion quite comprehensive and appropriate. It would be interesting of course to test other much better studied members of relaxins and related receptors for associations in this database. I would imagine that RLN2/RXFP1 mutations/variations might show associations with some vascular, fibrotic, or female reproductive phenotypes. Or, test SNPs in INSL3/RXFP2 genes, involved in testicular descent, for association with such clear-cut male phenotype as cryptorchidism.

The only thing I could not find is Table 1 (p.8, line 16) describing study demographics. Also, please correct “RXFP4” on page 2, line 13.

Reviewer #2: Dear Editor,

I have carefully reviewed the manuscript by Dr. Win Lee Edwin Wong entitled “Investigating the role of the relaxin-3/RXFP3 system in neuropsychiatric disorders and metabolic phenotypes: a candidate gene approach.”

In this manuscript the authors have used a large UK Biobank data analysis using single nucleotide polymorphisms (SNPs) pertinent to relaxin-3 signaling as potential genetic risk variants in association with depression, atypical depression, anxiety, and metabolic syndrome in a white UK population.

This study was comprehensively designed and conducted well. The study was reasonably well-powered, included clearly defined phenotype and case selection criteria, and data underwent rigid regression analysis using different complimentary population study based arithmetic methods.

The authors did not find a significant relationship between any of the candidate SNPs pertinent to relaxin-3 signaling (RLN3, RXFP3, RXFP4, RXFP1, RLN2) and depression, anxiety, or metabolic disorders investigated. They highlighted that despite a good rationale, candidate gene approaches have limitations when applied to genetic research into causes for neuropsychiatric disorders.

This is the first human study correlating relaxin-3 signaling system with depression, atypical depression, anxiety, and metabolic syndrome neuropsychiatric disorders and addresses an important knowledge gap in relaxin-related peptide/receptor research. This reviewer recommends this well-conducted study to be published in its present form.

6. PLOS authors have the option to publish the peer review history of their article (what does this mean?). If published, this will include your full peer review and any attached files.

Reviewer #1: No

Reviewer #2: No

---

## [Editor Report · Acceptance letter]

30 Oct 2023

PONE-D-23-28052 

Investigating the role of the relaxin-3/RXFP3 system in neuropsychiatric disorders and metabolic phenotypes: a candidate gene approach 

Dear Dr. Wong:

I'm pleased to inform you that your manuscript has been deemed suitable for publication in PLOS ONE. Congratulations! Your manuscript is now with our production department. 

Kind regards, 

on behalf of

Dr. Irina U Agoulnik 

Academic Editor

PLOS ONE